# Enhancing the Properties of Water-Soluble Copolymer Nanocomposites by Controlling the Layer Silicate Load and Exfoliated Nanolayers Adsorbed on Polymer Chains

**DOI:** 10.3390/polym15061413

**Published:** 2023-03-12

**Authors:** Dongyin Wang, Changfeng Chen, Xiaojuan Hu, Fei Ju, Yangchuan Ke

**Affiliations:** China National Petroleum Cooperation Key Laboratory of Nano-Chemistry, College of Science, China University of Petroleum, Beijing 102249, China

**Keywords:** copolymer nanocomposite, controlling exfoliated O-MMt nanolayer, high-temperature resistance, salt resistance, enhanced oil recovery

## Abstract

Novel polymer nanocomposites of methacryloyloxy ethyl dimethyl hexadecyl ammonium bromide-modified montmorillonite (O-MMt) with acrylamide/sodium *p*-styrene sulfonate/methacryloyloxy ethyl dimethyl hexadecyl ammonium bromide (ASD/O-MMt) were synthesized via in situ polymerization. The molecular structures of the synthesized materials were confirmed using Fourier-transform infrared and ^1^H-nuclear magnetic resonance spectroscopy. X-ray diffractometry and transmission electron microscopy revealed well-exfoliated and dispersed nanolayers in the polymer matrix, and scanning electron microscopy images revealed that the well-exfoliated nanolayers were strongly adsorbed on the polymer chains. The O-MMt intermediate load was optimized to 1.0%, and the exfoliated nanolayers with strongly adsorbed chains were controlled. The properties of the ASD/O-MMt copolymer nanocomposite, such as its resistance to high temperature, salt, and shear, were significantly enhanced compared with those obtained under other silicate loads. ASD/1.0 wt% O-MMt enhanced oil recovery by 10.5% because the presence of well-exfoliated and dispersed nanolayers improved the comprehensive properties of the nanocomposite. The large surface area, high aspect ratio, abundant active hydroxyl groups, and charge of the exfoliated O-MMt nanolayer also provided high reactivity and facilitated strong adsorption onto the polymer chains, thereby endowing the resulting nanocomposites with outstanding properties. Thus, the as-prepared polymer nanocomposites demonstrate significant potential for oil-recovery applications.

## 1. Introduction

Petroleum is an important energy source and crucial raw material for chemicals; thus, its worldwide demand is expected to continually increase. Because of the low efficiency of the traditional primary and secondary oil recovery methods, the need for enhanced oil recovery (EOR) alternatives has become a major priority [1,2,3]. In recent years, water-soluble polymer nanocomposites are increasingly being used in EOR [4,5,6,7]. Montmorillonite (MMt) is a common inorganic phase in the synthesis of water-soluble polymer nanocomposites, and its nano-effects have received considerable research attention [8,9,10,11,12,13,14,15,16,17]. MMt is a natural two-dimensional layered silicate material whose crystal structure is composed of two sheets of silicon–oxygen tetrahedra sandwiched between sheets of aluminum–oxygen octahedra [18,19,20]. The surface of a MMt layer has ample hydroxyl groups and negative charges, and abundant exchangeable ions exist between layers [21,22]. Therefore, MMt can be easily modified via layer intercalation to enlarge the layer–layer distance and reduce the interlayer interaction force. This enables reactive monomers to fill into the nano-layers and polymerize in the interlayer. The heat generated in the polymerization process causes the continuous expansion of the nano-slice spacing until an exfoliate state is reached. Exfoliated MMt layers have been prepared in this manner [23,24,25]. MMts with this structure are defined as MMt nanostructure intermediates (MMt NIs). Thus, polymer nanocomposites with well-exfoliated nanolayers can be prepared to enhance the comprehensive performance of the virgin polymer. Copolymers possessing strongly adsorbed exfoliated nanolayers, as well as high aspect ratios and large surface areas, disperse well in polymer matrices, where they exhibit excellent nano-effects [26,27,28,29]. Polymer nanocomposites with well-exfoliated MMt nanolayers have attracted significant research attention. Exfoliated clay layers improve the mechanical properties of clay/styrene–butadiene rubber nanocomposites [30]. Poly(ethylene oxide)/exfoliated-MMt nanocomposites demonstrate excellent thermal and mechanical properties [31], and poly(ethylene terephthalate)/exfoliated clay nanocomposites exhibit outstanding mechanical, thermal, and barrier properties [32]. The tunable expansion and plugging performance of polymer nanocomposite microspheres have also been reported [33]. Therefore, well-exfoliated MMt layers can impart unique properties to polymer nanocomposites.

Polymer nanocomposites with enhanced performance may have potential applications in oil recovery. The development of EOR methods has recently emerged as a strong research priority [1,2,3,34,35,36]. Polyacrylamide (PAM) and partially hydrolyzed polyacrylamide (HPAM), as water-soluble polymer (WSP) materials, have been mainly used to achieve mobility control in polymer flooding for EOR for the last several years [4,6,7,37]. WSP materials increase the viscosity of the displacing fluid and decrease the water/oil mobility ratio, thereby enhancing the displacement efficiency. However, PAM and HPAM are poorly stable at temperatures above 60 °C, and their solution viscosities decrease rapidly owing to the facile hydrolysis of their –CONH_2_ groups [38]. Under highly saline conditions, metal ions can impede PAM post-hydrolysis, leading to the formation of curled macromolecular chains [39]. In addition, a high shear rate decreases viscosity. Consequently, WSP materials become less effective in the oil flooding process under harsh reservoir conditions.

The applications of polymer–clay nanocomposites to the EOR process have been investigated. Sodium MMt (Na-MMt), which possesses a high swelling degree and cation-exchange capacity [40,41], has been introduced into polymers to improve their mechanical, thermal, and barrier properties. A modifier can be used as an intercalator to produce MMt NIs such that in situ copolymerization with suitable monomers yields exfoliated layers that can improve material properties by establishing strong interfacial interactions between the dispersed layers and polymer matrix. The optimized modifier reinforces the copolymer backbone [42]. Furthermore, methacryloyloxy ethyl dimethyl hexadecyl ammonium bromide (DMB), which is a quaternary ammonium compound with a terminal double bond, acts as an intercalator to Na-MMt, forming organic MMt (O-MMt) [43].

The sodium *p*-styrenesulfonate (SSS) monomer can be used to improve the temperature resistance of PAM because it exhibits good heat resistance at temperatures above 300 °C and high salinity resistance [44]. Poly(vinyl acetate-dibutyl maleate-acyclic acid-sodium *p*-styrene sulfonate) copolymer emulsions have high thermal resistance, while SSS displays suitable salt-tolerant characteristics [45]. β-Cyclodextrin (β-CD) is an environmentally friendly biodegradable macromolecule with good thermal stability. It has seven glucose units and abundant surface hydroxyl groups that facilitate chemical modification. β-CD-modified polymers have been used in EOR applications [46,47,48,49]. The introduction of β-CD improves the temperature resistance, salt tolerance, and anti-shearing performance of a polymer, thereby enabling its potential use in EOR applications in complex reservoirs. Liu et al. utilized the cavity of β-CD to prepare an acrylamide (AM)–β-CD copolymer exhibiting higher resistance to temperature, salt, and shear than PAM [50]. Polymer nanocomposites of β-CD-functionalized PAM have been applied to EOR [51], and polymer materials modified with β-CD have been used for EOR in high-temperature, high-mineralization oilfields [52]. In addition, β-CD modified with two 2-isocyanatoethyl acrylate (AOI) molecules results in β-CD-AOI_2_, which can be used to crosslink monomers and prepare a polymer gel exhibiting excellent performance [53].

In this study, we designed a tunable MMt NI using intercalation nanochemistry principles [54]. The selection of an appropriate modifier led to a MMt NI with a controllable interlayer distance; this intermediate underwent in situ copolymerization to produce well-exfoliated O-MMt active nanolayers that were uniformly dispersed in the polymer matrix. Thereafter, water-soluble copolymer nanocomposites (ASD/O-MMt) exhibiting the unique nano-effects of the exfoliated O-MMt nanolayers were synthesized. Depending on the O-MMt content and degree of exfoliation, these nanolayers strongly adsorbed onto polymer chains to yield polymer nanocomposites with excellent properties. Systematic analyses of the structure, properties, and solution flooding performance of these polymer nanocomposites were performed to optimize the nanomaterial composition. ASD/1.0 wt% O-MMt exhibited excellent properties because the exfoliated layers of the inorganic clay simultaneously exerted a positive effect on enhancing the final nanocomposite performance and a negative effect on increasing the molecular mass of the barrier polymer. The as-prepared optimized sample demonstrated potential applicability to the EOR process under high-temperature and high-salt reservoir conditions.

## 2. Experimental

### 2.1. Materials

Unmodified Na-MMt with a cationic exchange capacity of 100 mmol/100 g was supplied by Huai An Saibei Technology Co., Ltd., Zhangjiakou, China. Sodium chloride (NaCl, 98%), sodium hydroxide (NaOH, 96%), ammonium persulfate ((NH_4_)_2_S_2_O_8_, 98%), sodium hydrogen sulfite (NaHSO_3_, 98%), and absolute ethanol (CH_3_CH_2_OH, 99.7%) were purchased from the Tianjin Fuchen Fine Chemical Research Institute, Tianjin, China. AM (98%), acetone (CH_3_COCH_3_, 99.5%), and *N,N*-dimethylformamide (DMF, 99.5%) were purchased from the Tianjin Guangfu Fine Chemical Research Institute, Tianjin, China. SSS (98%) was obtained from Shanghai Macklin Biochemical Co., Ltd., Shanghai, China. Methacrylic acid 2-(dimethylamino)ethyl ester (stabilized with MEHQ, DMA, >98.5%) was supplied by TCI Industry Co., Ltd., Shanghai, China. 1-Bromohexadecane (97%), β-CD (98%), AOI (stabilized with BHT, >98.0%), and stannous octoate (95%) were purchased from Aladdin Biochemical Co., Ltd., Shanghai, China. All reagents were of analytical grade, and deionized water was used in all experiments.

### 2.2. Methods

#### 2.2.1. Preparation of DMB and β-CD-AOI_2_

1-Bromohexadecane (20.15 g, 0.066 mol) was added to a stirred solution of DMA (10.38 g, 0.066 mol) in 100 mL of anhydrous ethanol. Thereafter, the mixture was heated to 50 °C by continuously stirring in a water bath for 24 h. After the reaction, the solvent was removed using a rotary evaporator. The residue was first recrystallized in acetone and then washed several times with acetone to obtain DMB as a white powder. A scheme depicting the preparation of DMB is shown in Figure 1a.

β-CD (10 g, 7.6 mmol) was dissolved in 50 mL of anhydrous DMF under nitrogen. Thereafter, AOI (2.14 g, 15.2 mmol) was added to this mixture and stirred vigorously, after which 80 μL of tin(II) 2-ethylhexanoate (a catalyst) was added to the solution. The mixture was stirred for 90 min at 25 °C under nitrogen. The temperature was increased to 50 °C, and the reaction was allowed to proceed for 8 h under continuous stirring. At the end of the reaction, most of the solvent was removed using a rotary evaporator. The residue was recrystallized via the addition of cold acetone and then washed several times with deionized water and acetone to remove the DMF and unreacted AOI. The product was dried at 45 °C for 48 h in a vacuum oven. β-CD-AOI_2_ was obtained as a white powder and characterized using Fourier-transform infrared (FTIR) spectroscopy. A schematic depicting the synthesis of β-CD-AOI_2_ is shown in Figure 1b.

#### 2.2.2. Preparation of the O-MMt Intermediate

Pristine Na-MMt (3.33 g) was ultrasonically dispersed in 50 mL of distilled water to prepare a homogeneous mixture; this mixture was then heated to 50 °C. Subsequently, 50 mL of the DMB solution (3.33 g of DMB) were slowly added dropwise to the aforementioned mixture under continuous stirring. The cation-exchange reaction was allowed to proceed for 18 h under nitrogen. Finally, the sample was collected via filtration and washed several times with distilled water until the AgNO_3_ titration test confirmed the absence of bromide ions in the solution. The collected sample was dried in a vacuum oven at 70 °C for 24 h, ground into a powder, and carefully stored under dry conditions for further use. A schematic of the synthesis of O-MMt is shown in Figure 2.

#### 2.2.3. Preparation of the ASD/O-MMt Nanocomposite

The ASD/O-MMt nanocomposite was synthesized via in situ polymerization. First, a mixture of 0–0.67 g of O-MMt, 23.00 g of AM, 6.67 g of SSS, 3.33 g of DMB, 0.33 g of β-CD-AOI_2_, and 100 g of deionized water was added to a 250 mL three-necked flask and continuously stirred for 1 h under nitrogen. Thereafter, the pH of this mixture was adjusted to 8–9 using 1.0 mol/L NaOH. Second, a certain amount of (NH_4_)_2_S_2_O_8_ and NaHSO_3_ (molar ratio of 1:1; 0.3 wt%) was added as an initiator to the reaction mixture with stirring under nitrogen. The mixture was allowed to react at 50 °C for 12 h, after which the obtained product was washed several times with absolute ethanol and water (volume ratio of 9:1) to remove residual monomers and initiator. Finally, the gel sample was cut into pieces and dried in vacuo at 75 °C for 24 h. The final product was ground, sieved through a 200-mesh screen (74-μm eye size), and referred to as “ASD/x wt% O-MMt” (x = 0.5, 1.0, or 2.0). The pure polymer was prepared using the aforementioned process in the absence of O-MMt and referred to as “ASD.” The synthetic scheme of the ASD/O-MMt nanocomposites is shown in Figure 3.

### 2.3. Measurements

#### 2.3.1. Characterization

FTIR spectra were obtained at 25 °C on an FTS-3000 spectrophotometer (American Digilab) in the wavenumber range of 400–4000 cm^−1^. ^1^H-nuclear magnetic resonance (NMR) experiments were performed on a Bruker ASCEND-400 NMR spectrometer using D_2_O as the solvent. Thermogravimetric analysis (TGA) was performed on a NETZSCH analyzer (STA409PC, Selbu, Germany); the samples were heated from 25 to 700 °C at a rate of 10 °C/min. The morphologies of the nanocomposites were examined via scanning electron microscopy (SEM; SU8010, Hitachi, Japan), and the energy dispersive X-ray spectroscopy (EDS) was used to characterize the element in nanocomposites. O-MMt nanolayer exfoliation and dispersion in the polymer matrix were observed via transmission electron microscopy (TEM; JEM-2100, Tokyo, Japan) at an accelerating voltage of 200 kV.

X-ray diffractometry (XRD) was performed on a Bruker D8 Advance X-ray diffractometer (Karlsruhe, Germany) with Ni-filtered Cu-Kα radiation (*λ* = 1.54 Å). The diffraction angle (2θ) was varied from 1.0° to 10° at a rate of 1°/min and a scanning interval of 0.02°. The distance between silicate layers was calculated from the XRD data using Bragg’s law:(1)2dsinθ=nλ
where *n*, *λ*, *d*, and *θ* represent the reflection order, X-ray wavelength, interplanar distance (nm), and diffraction angle (°), respectively; here, *n* = 1.

#### 2.3.2. Intrinsic Nanocomposite Viscosity

The intrinsic viscosity [*η*] of each nanocomposite was measured using a Ubbelohde capillary viscometer at 25 °C. Prior to the measurements, the nanocomposite was dissolved in a 1.0 mol/L NaCl solution and diluted with 1.0 mol/L NaCl solution. The flow time for each nanocomposite solution was measured in triplicate; the difference between flow times did not exceed 0.2 s. The viscosity-average molecular weight (*M_η_*, g/mol) of each sample was calculated using the Mark–Houwink equation:(2)$$\eta_{sp}=(t-t_{0})/t_{0}$$
(3)$$\eta_{r}=t/t_{0}$$
(4)$$\left[\eta \right]=H/c$$
(5)$$\left[\eta \right]=4.75\times 10^{-3}M_{\eta }^{0.80}$$
where *t* and *t*_0_ are the flow times of the nanocomposite solution and 1.0 mol/L NaCl solution (s), respectively; *η_sp_* and *η_r_* are the specific and relative viscosities, respectively; *H* is the y-intercept of the two corresponding fitted straight lines; and *c* is the concentration of the nanocomposite solution (g/mL).

#### 2.3.3. Viscosity

The viscosities of the nanocomposite solutions at different concentrations, temperatures, and salinities were measured using a Brookfield DV-II + Pro viscometer at a rotor speed of 60 rpm. Each sample’s shear resistance at 0–200 s^−1^ and dynamic rheological behavior at 0–10 Hz were evaluated using a HAAKE RS600 controlled rheometer.

#### 2.3.4. EOR Tests

EOR tests were performed to investigate the performance of the ASD/1.0 wt% O-MMt polymer nanocomposite for EOR. The temperature and back pressure were maintained at 50 °C and 1500 psi, respectively. The detailed experimental procedures were as follows:(1)The core was saturated with synthetic brine (5217 mg/L), after which crude oil was injected into the core until water ceased to flow out from the outlet.(2)Water flooding was conducted at a flow rate of 0.5 mL/min until the water cut reached 98%.(3)A polymer slug having an injection volume (PV) of 0.3 was injected into the core, and water flooding was continued until the water cut reached 98%.

An artificial sandstone core was employed for the experiments, and the basic experimental parameters are listed in Table 1. Table 2 lists the composition of the synthetic brine used in the experiments. The oil recovery ratio was calculated using the following equation [55]:(6)$$EOR=E_{T}-E_{W}$$
where *EOR* is the enhanced oil recovery obtained using the polymer solution (%); *E_T_* is the total oil recovery from the total flooding process (%); and *E_W_* is the oil recovery from the water flooding process (%).

## 3. Results and Discussion

### 3.1. FTIR Spectroscopy

The chemical structure of β-CD-AOI_2_ was identified from the FTIR spectra of β-CD and β-CD-AOI_2_. As shown in Figure 4I, the new characteristic IR bands at 1705 and 1595 cm^−1^, which are absent in the spectrum of β-CD, correspond to the stretching vibrations of secondary amides, thereby confirming the successful grafting of AOI onto β-CD [8].

To obtain more information on the structure of the copolymer nanocomposites, we obtained the FTIR spectra of pristine Na-MMt, O-MMt, ASD, and ASD/1.0 wt% O-MMt (Figure 4II). The characteristic band at 3632 cm^−1^ observed for pristine Na-MMt corresponds to the O–H stretching vibrations of Al–OH and Si–OH in the silicate layers. The broad band at 3450 cm^−1^ and the sharp band at 1630 cm^−1^ are related to hydroxyl stretching and H–O–H bending vibrations, thus confirming the presence of free and interlayer water molecules on the MMt clay. The bands at 1043, 524, and 466 cm^−1^ are associated with Si–O–Si stretching vibrations, and the peak at 781 cm^−1^ can be assigned to Si–O–Al stretching vibrations. The peaks at 2929 and 2858 cm^−1^ in the spectrum of O-MMt correspond to the symmetric -CH_2_ and asymmetric -CH_2_ telescopic vibrations of the organic long-chain molecule, respectively. The peaks at 1620, 1740, and 1120 cm^−1^ are attributed to telescopic C=C vibrations, symmetric and asymmetric -C=O stretching vibrations, and -C-O stretching vibrations in DMB molecules, respectively, thus suggesting the successful synthesis of DMB-modified MMt. The broad absorption band at 3430 cm^−1^ in the spectrum of the ASD copolymer is assigned to the N–H stretching vibrations of -CONH_2_. The band at 1415 cm^−1^ is a characteristic benzene ring peak. The absorption peaks at 1136 and 1085 cm^−1^ are related to the antisymmetric and symmetric S=O stretching vibrations of -SO_3_^2−^, respectively, which confirm the successful synthesis of the AM/SSS/DMB copolymer. The additional peaks at 524 and 466 cm^−1^ in the FTIR spectrum of ASD/1.0 wt% O-MMt (Figure 4II(d)) are assigned to Si–O–Si bending vibrations, while the band at 781 cm^−1^ corresponds to Si–O–Al stretching vibrations in the silicate clay. These findings confirm the successful synthesis of the ASD/1.0 wt% O–MMt nanocomposite [33,56,57]. The FTIR spectra of ASD/0.5 wt% O-MMt and ASD/2.0 wt% O-MMt are presented in Appendix A.

### 3.2. ^1^H-NMR Spectroscopy

To further confirm that the ASD copolymer had been successfully synthesized, we obtained the ^1^H-NMR spectrum of ASD using D_2_O as the solvent (Figure 5). The strong peak at 4.68 ppm corresponds to residual H_2_O in D_2_O. The peaks at 1.75–1.35 (1, 8) and 2.07–1.95 ppm (2) correspond to the -CH_2_- and -CH- protons of the polymer backbone, respectively. The singlet at 2.22 ppm (9) corresponds to the -CH_3_ protons in DMB, and the peaks at 6.25–5.53 ppm (4, 5) are characteristic -CH_2_- and -CH proton peaks of the polymer backbone of SSS. The peaks at 7.68 (7) and 7.20 ppm (6) are attributable to benzene ring protons. The small peak at 6.89 ppm (3) corresponds to the -NH_2_ proton in AM, and the multiplets at 3.34 (12) and 3.72 ppm (11) correspond to the -CH_2_- and -CH_3_ protons in the long alkyl chain of DMB. The peak at 2.37 ppm (10) and multiplet at 3.56–3.50 ppm (13) are attributable to the -OCH_2_- and -CH_2_- protons that are bonded to DMB, while the triplet at 1.05 ppm (14) corresponds to the terminal -CH_3_ protons in the hydrophobic chain of DMB. Therefore, ^1^H-NMR analysis confirmed that the target product had been successfully prepared.

### 3.3. XRD

The layer spacing is usually determined using XRD. The diffraction peak position on the d_001_ surface of MMt can reflect the interlayer spacing (d) of MMt [58]. The XRD patterns of Na-MMt, O-MMt, ASD, ASD/0.5 wt% O-MMt, ASD/1.0 wt% O-MMt, and ASD/2.0 wt% O-MMt are shown in Figure 6. According to Figure 6a,b, Na-MMt exhibits a characteristic diffraction peak at 7.30°, while the diffraction peaks of O-MMt are observed at 4.50° and 2.25°. Based on Bragg’s law, O-MMt has layer spacings of 19.5 and 39.1 Å; hence, Na-MMt (initial layer spacing of 13.0 Å) was successfully modified with DMB and O-MMt was successfully prepared. Diffraction peaks disappear as the interlayer spacing of O-MMt expands in the exfoliated nanolayers. No characteristic diffraction peaks were detected for ASD/0.5 wt% O-MMt, ASD/1.0 wt% O-MMt, and ASD/2.0 wt% O-MMt (Figure 6c–f). This shows that the polymer chains entered into the interlayer of O-MMt during the exothermic polymerization process. This heat generation accelerates the exfoliation of O-MMt nanolayers dispersed in the polymer matrix. In addition, the dispersion and exfoliation degree of O-MMT sheets and the microstructure of the polymer matrix were confirmed via TEM analysis.

### 3.4. SEM

Figure 7 shows the SEM images of the prepared ASD polymer and ASD/1.0 wt% O-MMt. The network structure of the pure polymer is not very dense and is easily destroyed, as shown in Figure 7a,b; in addition, the polymer has a very fine and highly porous chain skeleton. ASD/1.0 wt% O-MMt exhibits a network morphology (Figure 7c,d) with interconnected skeletons and cavities. The EDS analysis of ASD/1.0 wt% O-MMt (Figure 7e) confirms the presence of C, N, O, and S from the polymer and Na, Mg, Al, and Si from the inorganic silicate nanosheets, indicating that the O-MMt additive was successfully incorporated into the ASD polymer. Compared with the pure polymer, ASD/1.0 wt% O-MMt has a considerably more compact network structure, with a stronger skeleton and fewer but smaller cavities. These features are ascribable to the surface of the exfoliated O-MMt layers having abundant and highly reactive groups, enabling the strong adsorption of polymer molecular chains onto the material surface; these layers act as a nanobarrier that protects the polymer. Therefore, a coarse polymer skeleton, less porous structure, and strong network structure are formed. Furthermore, Figure 7d shows that the exfoliated O-MMt nanolayers are strongly adsorbed on the polymer chain surface because the abundant hydroxyl (Al-OH and Si-OH) and amino groups on the exfoliated O-MMt layers participate in intramolecular and intermolecular hydrogen bonding, resulting in the formation of a thicker and stronger network structure. These results indicate that ASD/1.0 wt% O-MMt exhibits better mechanical properties than ASD and has good comprehensive properties that are important for EOR from high-temperature and high-salinity reservoirs.

### 3.5. TEM

Well-exfoliated and well-dispersed O-MMt nanolayers in the polymer matrix can significantly improve the properties of a polymer nanocomposite, such as its viscosity, temperature resistance, salt resistance, shear resistance, and viscoelasticity. To obtain clearer and more reliable information on the exfoliated O-MMt nanolayers dispersed in the nanocomposite, we captured TEM images of ASD/1.0 wt% O-MMt (Figure 8). During TEM, the electron beam penetrates the sample, and the black regions in the image provide estimates of the degrees of electron beam penetration.

Various parts of the samples showed different nanolayer thicknesses. The O-MMt nanolayers appear as dark lines in the TEM micrographs, in which several exfoliated O-MMt nanolayers are observed, as shown in Figure 8a,b. The red arrows in the images indicate dispersions of exfoliated O-MMt nanolayers. Numerous single O-MMt layers are observed in Figure 8c,d. The exfoliated nanolayers were uniformly distributed in the polymer matrix and exerted a significant nano-effect on the performance of the ASD/1.0 wt% O-MMt nanocomposite. The TEM images of ASD, ASD/0.5 wt% O-MMt, and ASD/2.0 wt% O-MMt are presented in Appendix A.

In addition to strong adsorption ascribable to the reactive groups on the surfaces of the O-MMt layers, ion–dipole interactions between sulfonic groups and exchangeable cations, as well as those between quaternary ammonium groups and negative charges on the O-MMt sheets, accelerate the exfoliation of O-MMt. In particular, polymer reorganization and the presence of functional groups (e.g., hydroxyl, amido, and quaternary ammonium groups) facilitate the exfoliation of the O-MMt layers. These well-exfoliated and well-dissipated O-MMt nanolayers can act as nano barriers and strongly adsorb onto the polymer chains, thereby slowing down the thermal decomposition rate of the nanocomposite. Consequently, improved thermal stability can be expected from the O-MMt-loaded sample. Thus, ASD/1.0 wt% O-MMt shows high temperature stability toward water-soluble fluids, which is beneficial for EOR.

### 3.6. TGA

Figure 9 shows the TGA curves of ASD, ASD/0.5 wt% O-MMt, ASD/1.0 wt% O-MMt, and ASD/2.0 wt% O-MMt. The maximum degradation temperature of ASD/1.0 wt% O-MMt was 408 °C, while that of ASD was 392 °C (Figure 9I). Therefore, the introduction of O-MMt improved the heat resistance of the polymer nanocomposites. The maximum degradation temperature of the nanocomposite was 16 °C higher than that of ASD at an O-MMt content of 1 wt%, but only 6 °C higher at an O-MMt content of 2 wt%. Therefore, a moderate amount of O-MMt (1 wt%) significantly improved the temperature resistance of the nanocomposite because the O-MMt layers were not only well-exfoliated and well-dispersed in the polymer matrix but also strongly adsorbed onto the polymer, where they acted as nano barriers and significantly improved the thermal stability of the material. The residual mass of ASD/1.0 wt% O-MMt was 28.8% at 700 °C, whereas those of ASD/0.5 wt% O-MMt and ASD/2.0 wt% O-MMt were 25.9% and 28.2%, respectively; the residual mass of ASD was 24%. In other words, ASD/1.0 wt% O-MMt lost 4.8% more weight than the pure polymer at 700 °C. ASD/1.0 wt% O-MMt exhibited a higher residual mass and better thermal stability because the O-MMt layers protected the polymer matrix against thermal decomposition, owing to the barrier effect originating from the high surface area and aspect ratio of the O-MMt nanolayers [59]. As depicted in Figure 9II, ASD/1.0 wt% O-MMt has the highest temperature at the maximum thermal decomposition rate. Therefore, thermally stable polymer nanocomposites with well-exfoliated and well-dispersed O-MMt nanolayers (ASD/1.0 wt% O-MMt) are potentially useful in EOR applications involving high-temperature reservoirs.

### 3.7. Intrinsic Viscosity and Viscosity-Average Molecular Weight

The viscosity of a polymer depends on its concentration and size at a particular temperature in a specific solvent. The property [*η*] is a physical quantity that reflects the nature of the molecule. The [*η*] and *M_η_* of a polymer can be obtained via the dilution method. A plot of *η_sp_*/*c* or ln*η_r_*/*c* against concentration yields a straight line, and [*η*] can be determined by extrapolating this line to the y-intercept. The *M_η_* of each sample was calculated using the Mark–Houwink equation. The plot of *η_sp_*/*c* or (ln*η_r_*/*c*) versus ASD concentration is shown in Figure 10, and the details of the ASD/O-MMt nanocomposites are listed in Table 3. The relatively high *M_η_* of ASD/1.0 wt% O-MMt could be attributed to the well-exfoliated and well-dispersed O-MMt nanolayers. The relationship between *η_sp_*/*c* or *lnη_r_*/*c* and the ASD/0.5 wt% O-MMt, ASD/1.0 wt% O-MMt, and ASD/2.0 wt% O-MMt concentrations are presented in Appendix A. The high viscosity of the polymer nanocomposite (ASD/1.0 wt% O-MMt) is beneficial for improving the sweep area in a high-temperature reservoir for EOR.

### 3.8. Viscosity

Polymer nanocomposite samples with various concentrations were prepared by dissolving the appropriate amount of the polymer in 100 mL of distilled water, after which the apparent viscosity of each sample solution was measured at 25 °C. Figure 11 shows that the apparent viscosity increases with an increasing solution concentration. The apparent viscosity of ASD/1.0 wt% O-MMt changed from 18.8 to 208.9 mPa·s as the concentration was increased from 1 to 11 g/L. The apparent viscosities of ASD, ASD/0.5 wt% O-MMt, and ASD/2.0 wt% O-MMt were 145.9, 112.3, and 132.9, respectively, at 11 g/L. Therefore, ASD/1.0 wt% O-MMt exhibited the highest apparent viscosity among the analyzed samples under the same conditions. Strong interactions between the well-exfoliated and well-dispersed O-MMt nanolayers and the polymer, as explained previously, significantly increased the viscosity of the WSP. This tackifying property increased the sweep area of a WSP, which is beneficial for EOR.

### 3.9. Temperature Resistance

High temperatures can significantly damage polymer chains by destroying intermolecular physical interactions and accelerating polymer decomposition. Therefore, the apparent viscosity decreases with increasing temperature. Sample solutions were prepared at a concentration of 5 g/L, and the effects of temperature on the viscosities of the ASD and ASD/O-MMt solutions were investigated (Figure 12). The viscosities of the ASD and ASD/O-MMt solutions were found to depend on the temperature, with the viscosity decreasing as the temperature increased from 25 to 95 °C. Apparent viscosities of 11.1, 6.0, 23.5, and 7.9 mP·s were determined for the ASD, ASD/0.5 wt% O-MMt, ASD/1.0 wt% O-MMt, and ASD/2.0 wt% O-MMt solutions, respectively, at 95 °C. The apparent viscosity of ASD/1.0 wt% O-MMt was consistently higher than those of the other samples at a given temperature, which is indicative of its superior temperature resistance. Polymer nanocomposite degradation was retarded because the well-exfoliated and well-dispersed O-MMt nanolayers strongly adsorbed onto the surfaces of the polymer chains and acted as nano barriers. Moreover, strong interactions between the O-MMt nanolayers and the polymer restricted polymer chain movement at high temperatures and improved the temperature resistance of the nanocomposite, thereby contributing to EOR from high-temperature reservoirs.

### 3.10. Salt Resistance

Salt resistance is an important index for the use of an ASD/O-MMt nanocomposite in EOR applications. Therefore, we investigated the effect of different NaCl concentrations on the apparent viscosities of ASD and ASD/O-MMt nanocomposite solutions (5 g/L) at 25 °C (Figure 13). The apparent viscosity of the ASD solution decreased from 48.2 to 5.8 mPa·s as the NaCl concentration increased from 0 to 10 g/L, with apparent viscosities decreasing from 13.5, 87.4, and 21.7 mPa·s to 2.0, 9.2, and 3.8 mPa·s for ASD/0.5 wt% O-MMt, ASD/1.0 wt% O-MMt, and ASD/2.0 wt% O-MMt, respectively. The apparent viscosity of ASD/1.0 wt% O-MMt was consistently higher than those of the other composites. The strongly adsorbed O-MMt nanolayers on the polymer surface prevented inorganic salt ions from affecting the polymer chains; hence, the polymer chains remained fully extended in aqueous solution, thereby maintaining its viscosity. The barrier effect of the exfoliated O-MMt nanolayers improved the salt resistance of the solution; however, adsorption on the polymer surface was poor in the absence of well-exfoliated and well-dispersed O-MMt nanolayers; under these conditions, no nanolayers were available to act as nano barriers, resulting in polymer degradation and a decrease in apparent viscosity. Electrostatic repulsion in the polymer chain decreased when the Na^+^ and -SO_3_^2−^ concentration changes were neutralized, leading to the curling of the macromolecular chain and an associated decrease in viscosity. ASD/1.0 wt% O-MMt exhibited a higher viscosity retention rate than ASD in the same solvent. These results indicate that ASD/1.0 wt% O-MMt can be used for EOR in high-salinity reservoirs.

### 3.11. Shear Resistance

Shear forces from the pump, pipeline, wellbore, bullet hole, and porous media can significantly reduce the viscosity of the polymer solution and affect oil recovery during the injection and displacement processes. Figure 14 shows the apparent viscosities of ASD, ASD/0.5 wt% O-MMt, ASD/1.0 wt% O-MMt, and ASD/2.0 wt% O-MMt at various shear rates. ASD/1.0 wt% O-MMt displayed a higher apparent viscosity than ASD, ASD/0.5 wt% O-MMt, and ASD/2.0 wt% O-MMt between 0 and 200 s^−1^. As the shear rate increased from 0 to 100 s^−1^, the apparent viscosity of ASD/1.0 wt% O-MMt decreased from 176.9 to 44.13 mPa·s, that of ASD/2.0 wt% O-MMt decreased from 108.7 to 38.93 mPa·s, that of ASD decreased from 118.3 to 24.45 mPa·s, and that of ASD/0.5 wt% O-MMt decreased from 96.0 to 2.15 mPa·s. The apparent viscosity of ASD/1.0 wt% O-MMt remained higher than those of ASD, ASD/0.5 wt% O-MMt, and ASD/2.0 wt% O-MMt at each shear rate tested in the 100–200 s^−1^ range; hence, ASD/1.0 wt% O-MMt was more shear-resistant than the other polymer nanocomposites. Because the O-MMt nanolayers of ASD/1.0 wt% O-MMt strongly adsorbed onto the polymer surface, this polymer nanocomposite had a stronger framework and coarser polymer chains, showing better mechanical properties than the other samples. The shear resistance of ASD/2.0 wt% O-MMt was greater than that of ASD due to the friction resistance of the MMt layer. These properties contribute to the improved shear resistance of the nanocomposite. Improved shear resistance renders the ASD/1.0 wt% O-MMt polymer nanocomposite very useful for EOR.

### 3.12. Viscoelasticity

Viscoelastic fluids can enhance oil recovery; hence, we studied the dynamic rheological behavior of the polymer nanocomposites. Relationships between the *G*′ and *G*″ moduli and frequency are shown in Figure 15. ASD/1.0 wt% O-MMt displayed *G*′ and *G*″ values that are higher than those of ASD, ASD/0.5 wt% O-MMt, and ASD/2.0 wt% O-MMt, indicating better viscoelasticity. *G*′ and *G*″ crossover at *G*_c_, and the frequency corresponding to this point is the characteristic frequency (*f*_c_) of the material. *G*″ is higher than *G*′ for ASD/1.0 wt% O-MMt (Figure 15) at frequencies less than *f*_c_, where the viscidity of the polymer nanocomposite dominates, and the fluid exhibits liquid-like properties. By contrast, elasticity dominates at frequencies above *f*_c_, where *G*’ is higher than *G*″; the *G*′ value of the fluid rapidly increases with increasing frequency, and the fluid tends to exhibit elastic-fluid-like behavior, which increases the sweep area and promotes adsorption to oil surfaces on rocks, thereby enhancing oil recovery. The O-MMt nanolayers have ample hydroxyl groups, hydrogen bonds and ion–dipole interactions among the nanolayers, polymer chains, and oil surface, thus facilitating the removal of oil droplets from rocks, thereby yielding more oil.

### 3.13. EOR Experiments under Laboratory Conditions

Experiments were designed and conducted to evaluate the potential applications of ASD/1.0 wt% O-MMt for EOR. The results are shown in Figure 16. First, during water flooding, the injection pressure and oil recovery increased with the injection of water. The oil recovery of this first water flooding process was *E_W_*. Second, the injection pressure and oil recovery increased rapidly with polymer flooding. Third, in the subsequent water flooding process, the oil recovery reached a stable value (*E_T_*). However, the injection pressure first increased and then decreased with an increasing PV. Compared with the water flooding processes, the use of ASD/1.0 wt% O-MMt led to 10.5% higher oil recovery. The experimental parameters and results are listed in Table 4. The EOR test of ASD, ASD/0.5 wt% O-MMt, and ASD/2.0 wt% O-MMt are presented in Appendix A. The results indicate that ASD/1.0 wt% O-MMt has significant potential for application to EOR.

### 3.14. Proposed Adsorption Mechanism of the Exfoliated O-MMt Nanolayers

The well-exfoliated and well-dissipated O-MMt nanolayers strongly adsorb onto the surfaces of the polymer chains, thereby significantly improving the properties of the polymer nanocomposite. The mechanism for the adsorption of exfoliated O-MMt nanolayers is shown in Figure 17. The amino groups of AM form hydrogen bonds with the hydroxyl groups on the exfoliated O-MMt nanolayers. Meanwhile, hydronium ions (H_3_O^+^) act as bridging molecules by forming hydrogen bonds with the hydroxyl and N–H groups of AM. Because of its sulfonic groups, SSS strongly adsorbs onto the surfaces of the polymer chains through H_3_O^+^. Moreover, ionic bonds are formed between the ammonium centers (N^+^) of DMB molecules and the exfoliated O-MMt nanolayers. The terminal double bond of DMB polymerizes to form covalent bonds. Overall, polymer chains are strongly adsorbed onto exfoliated O-MMt nanolayers as a result of covalent, ionic, and hydrogen bonding; these strongly adsorbed nanolayers protect the polymer nanocomposite from damage. Our results reveal that ASD/1.0 wt% O-MMt, with well-exfoliated and well-dispersed O-MMt nanolayers, exhibits excellent temperature resistance, salt resistance, shear resistance, and viscoelasticity. Therefore, this polymer nanocomposite is potentially useful for EOR applications [17,60].

## 4. Conclusions

ASD/O-MMt nanocomposites with well-exfoliated and well-dispersed O-MMt nanolayers were successfully synthesized via the in situ polymerization of AM, DMB, SSS, and DMB-modified O-MMt. FTIR and ^1^H-NMR spectroscopy confirmed that ASD/1.0 wt% O-MMt had been successfully synthesized. XRD, TEM, and SEM analyses revealed that the O-MMt nanolayers in the polymer matrix were well-exfoliated and well-dispersed, indicating strong adsorption between the O-MMt nanolayers and polymer chains. Compared with other polymer nanocomposites, ASD/1.0 wt% O-MMt exhibited superior resistance to high temperature, salt, and shear and good viscoelasticity; it also enhanced oil recovery by 10.5%. The well-exfoliated and well-dispersed O-MMt nanolayers significantly improved the thermal stability of the ASD/O-MMt composites. The excellent performance of the nanocomposites was attributed to a strong adsorption between the O-MMt nanolayers and polymer chains, for which a plausible adsorption mechanism was proposed. All the results suggest that ASD/1.0 wt% O-MMt could potentially be applied to EOR in high-temperature and high-salinity reservoirs.

## Figures and Tables

**Figure 1 polymers-15-01413-f001:**
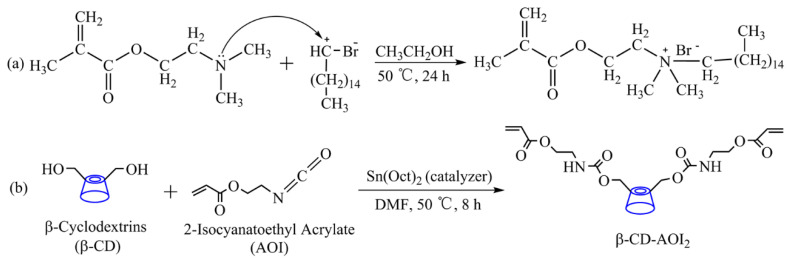
Synthesis of (**a**) DMB and (**b**) β-CD-AOI_2_.

**Figure 2 polymers-15-01413-f002:**
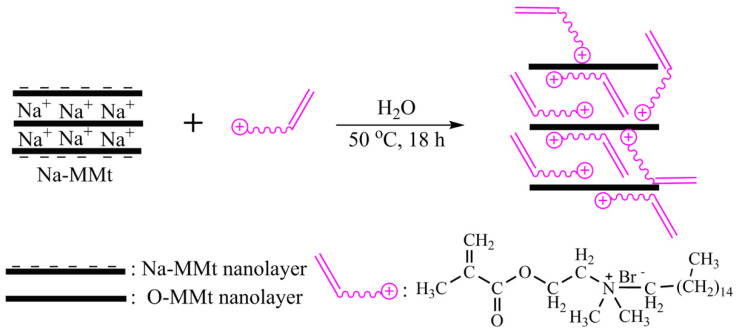
Preparation of the O-MMt intermediate.

**Figure 3 polymers-15-01413-f003:**
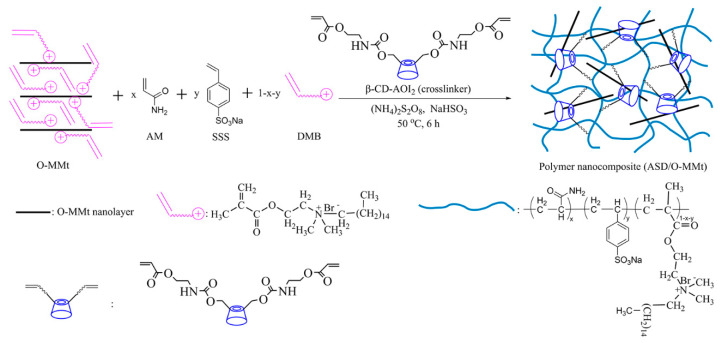
Synthesis of ASD/O-MMt nanocomposites.

**Figure 4 polymers-15-01413-f004:**
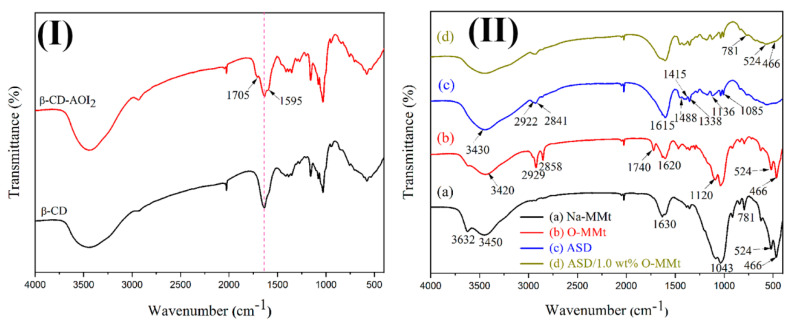
FTIR spectra of (**I**) β−CD and β−CD−AOI_2_ and (**II**) (**a**) pristine Na−MMt, (**b**) O−MMt, (**c**) ASD, and (**d**) ASD/1.0 wt% O−MMt.

**Figure 5 polymers-15-01413-f005:**
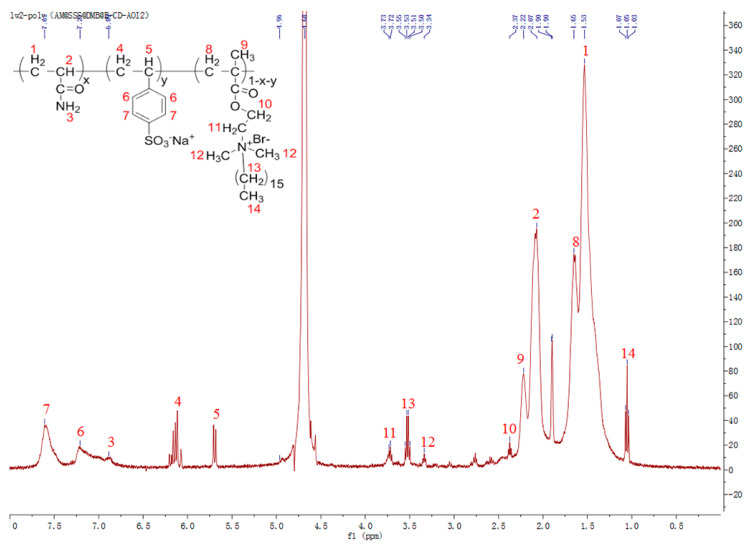
^1^H−NMR spectrum of ASD.

**Figure 6 polymers-15-01413-f006:**
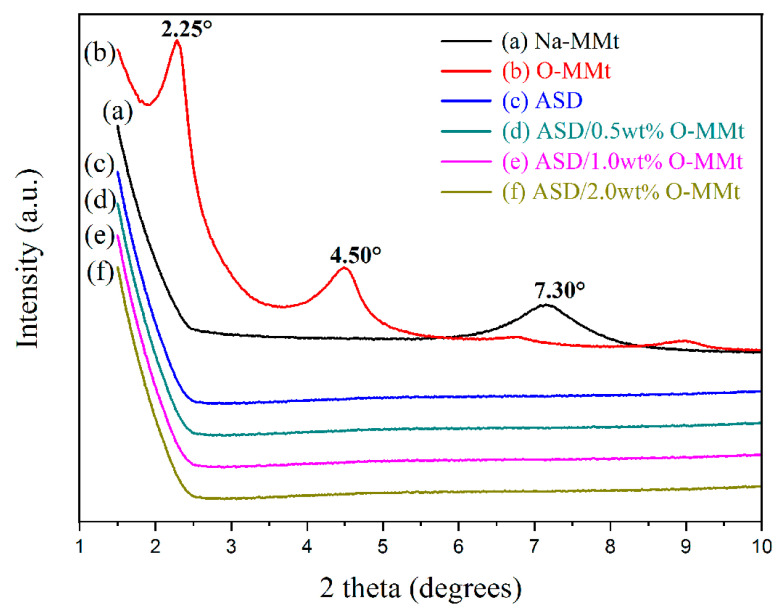
XRD patterns of (**a**) Na-MMt, (**b**) O-MMt, (**c**) ASD, (**d**) ASD/0.5 wt% O-MMt, (**e**) ASD/1.0 wt% O-MMt, and (**f**) ASD/2.0 wt% O-MMt.

**Figure 7 polymers-15-01413-f007:**
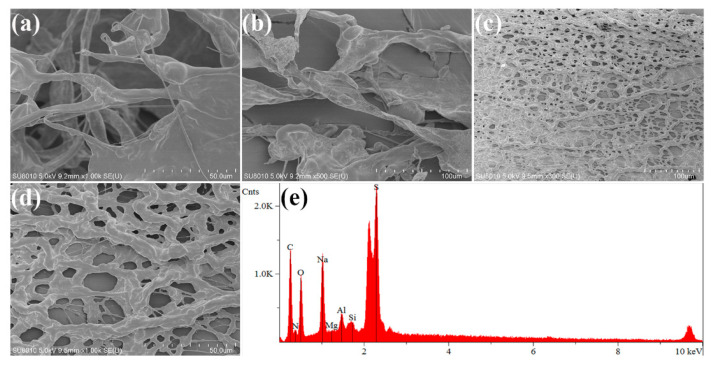
SEM images of (**a**,**b**) ASD and (**c**,**d**) ASD/1.0 wt% O-MMt and (**e**) EDS analysis of ASD/1.0 wt% O-MMt.

**Figure 8 polymers-15-01413-f008:**
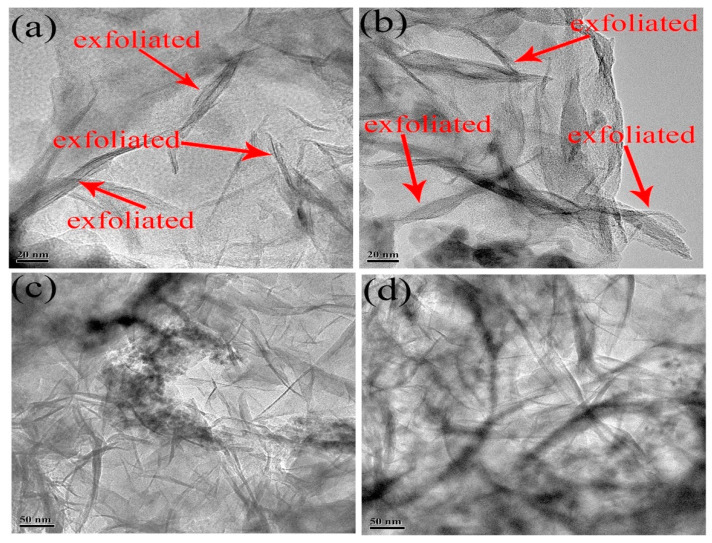
TEM images of (**a**–**d**) well-exfoliated and well-dispersed ASD/1.0 wt% O-MMt.

**Figure 9 polymers-15-01413-f009:**
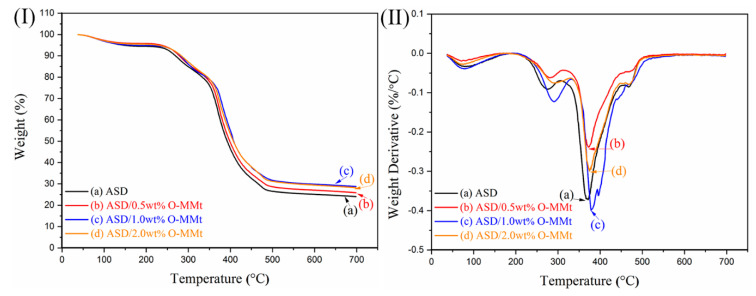
(**I**) TGA and (**II**) DTG curves of (**a**) ASD, (**b**) ASD/0.5 wt% O-MMt, (**c**) ASD/1.0 wt% O-MMt, and (**d**) ASD/2.0 wt% O-MMt.

**Figure 10 polymers-15-01413-f010:**
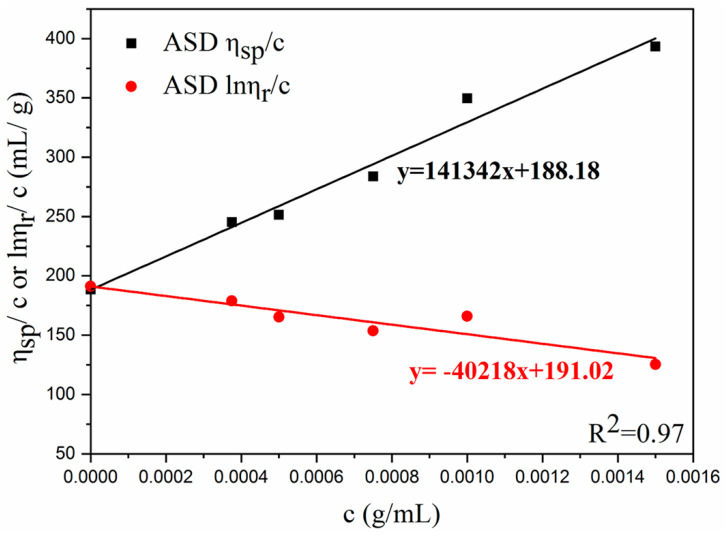
Relationship between *η_sp_*/*c* or ln*η_r_*/*c* and the ASD concentration.

**Figure 11 polymers-15-01413-f011:**
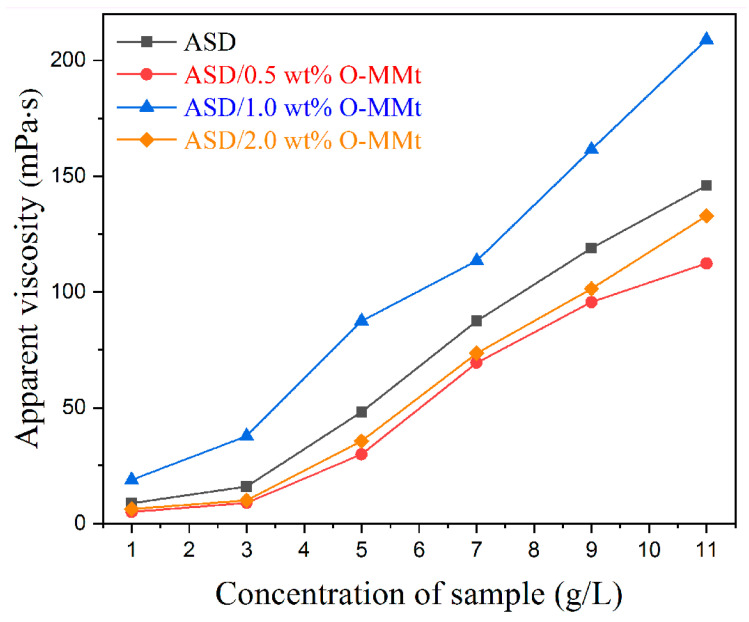
Apparent viscosity of the water-soluble polymer nanocomposites.

**Figure 12 polymers-15-01413-f012:**
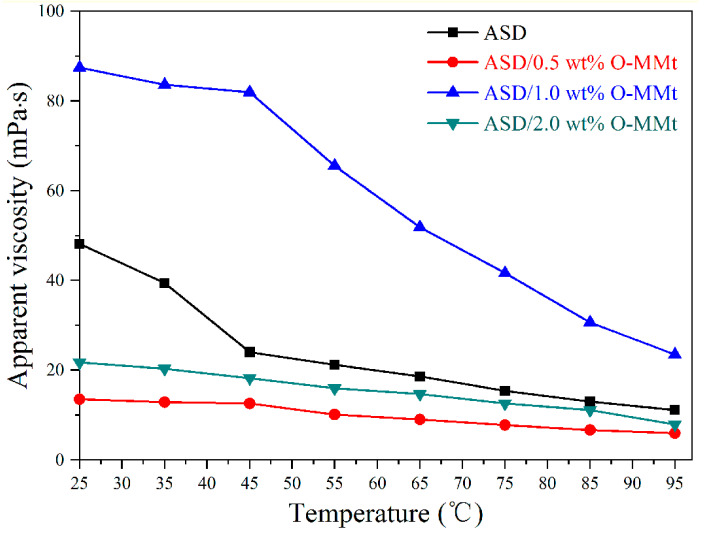
Relationships between apparent viscosity and temperature for ASD, ASD/0.5 wt% O-MMt, ASD/1.0 wt% O-MMt, and ASD/2.0 wt% O-MMt.

**Figure 13 polymers-15-01413-f013:**
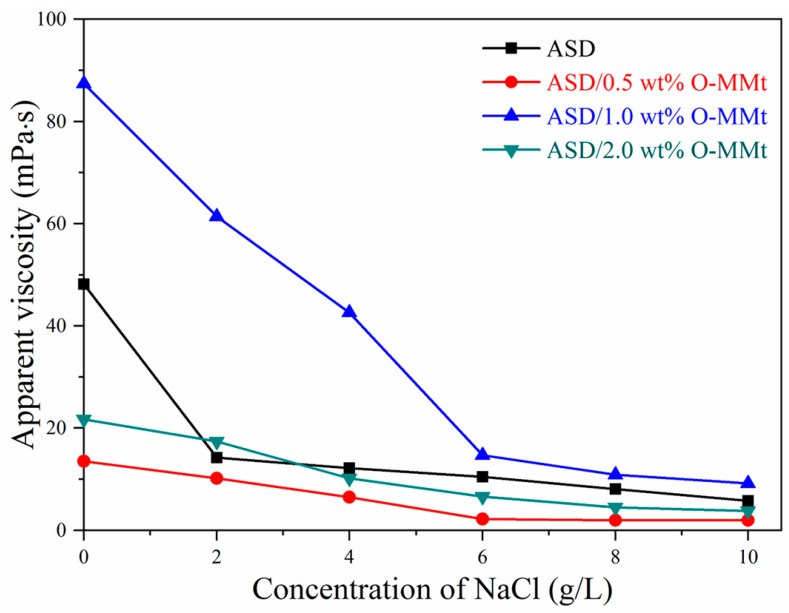
Relationships between the apparent viscosities of ASD, ASD/0.5 wt% O-MMt, ASD/1.0 wt% O-MMt, ASD/2.0 wt% O-MMt, and NaCl concentration.

**Figure 14 polymers-15-01413-f014:**
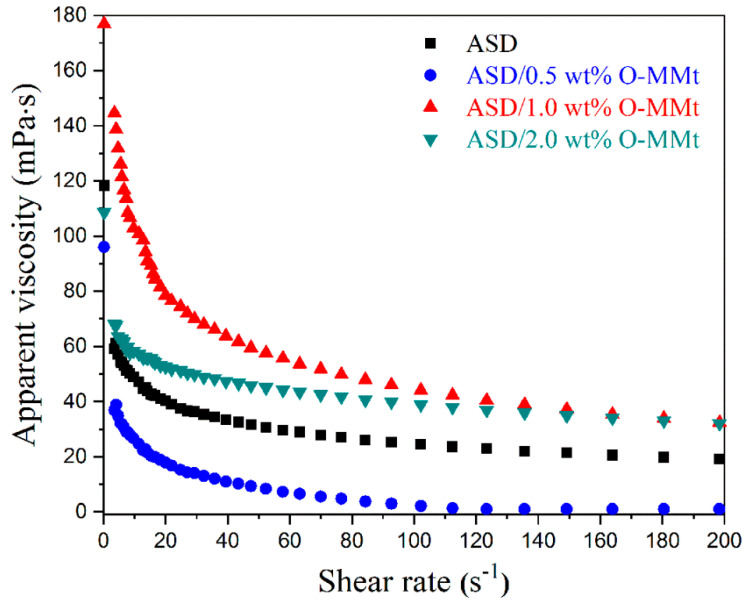
Effect of shear rate on the apparent viscosities of the nanocomposites (5 g/L) at 25 °C.

**Figure 15 polymers-15-01413-f015:**
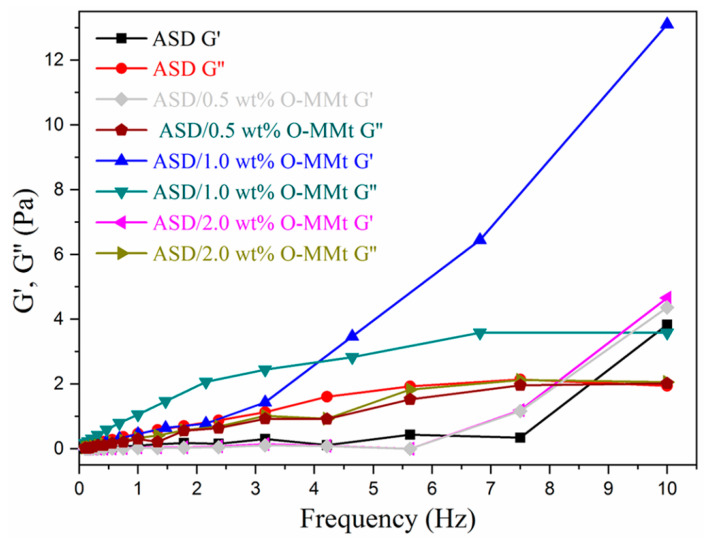
Elastic (G′) and viscous (G″) moduli as a function of frequency for ASD, ASD/0.5 wt% O-MMt, ASD/1.0 wt% O-MMt, and ASD/2.0 wt% O-MMt.

**Figure 16 polymers-15-01413-f016:**
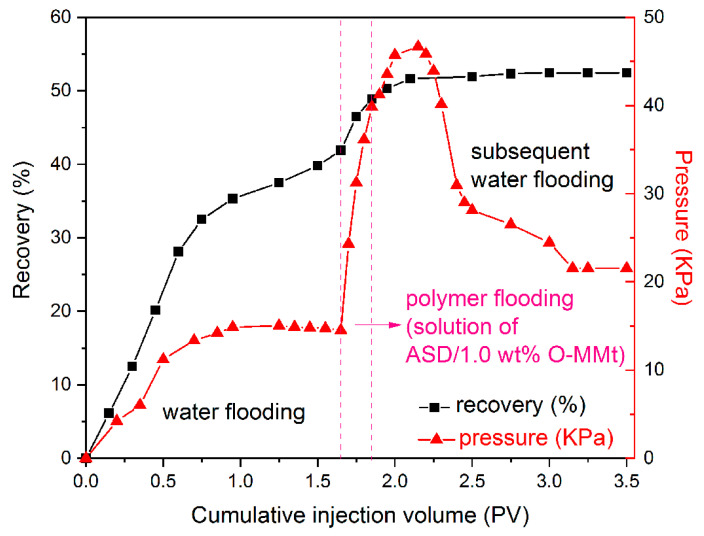
EOR experiments using ASD/1.0 wt% O-MMt under laboratory conditions.

**Figure 17 polymers-15-01413-f017:**
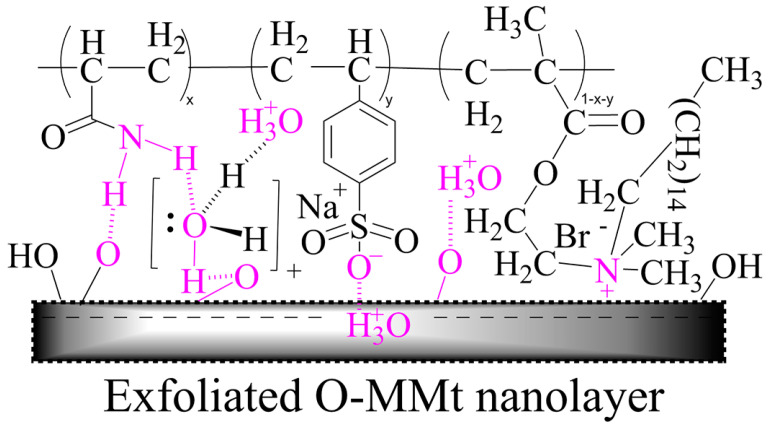
Proposed mechanism for exfoliated O-MMt nanolayer adsorption.

**Table 1 polymers-15-01413-t001:** Basic parameters of the artificial sandstone cores.

Core	Length (cm)	Diameter (cm)	Porosity (%)	Permeability (mD)
1#	7.76	2.53	22.13	1039

**Table 2 polymers-15-01413-t002:** Composition of the synthetic brine solution.

Composition	NaCl	KCl	CaCl_2_	MgCl_2_·5H_2_O	NaSO_4_	NaHCO_3_	Total
Concentration (mg/L)	2728	15	55	135	70	2214	5217

**Table 3 polymers-15-01413-t003:** Intrinsic viscosities [*η*] and viscosity-average molecular weights (*M_η_*) of the synthesized polymers.

Sample	O-MMt Load (wt%)	[*η*] (mL/g)	*Mη* × 10^5^ (g/mol)	Standard Deviations
ASD	0	189.60	5.64	1.39
ASD/0.5 wt% O-MMt	0.5	109.21	2.83	1.63
ASD/1.0 wt% O-MMt	1	295.06	9.81	1.27
ASD/2.0 wt% O-MMt	2	146.32	4.08	1.98

**Table 4 polymers-15-01413-t004:** EOR of ASD/1.0 wt% O-Mt.

Sample	Concentration (mg/L)	*E_W_* (%)	*E_T_* (%)	EOR (%)
ASD/1.0 wt% O-MMt	1000	41.9	52.4	10.5

## Data Availability

The data presented in this study are available on request from the corresponding author.

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
