# Peer review of "Enhancing the Properties of Water-Soluble Copolymer Nanocomposites by Controlling the Layer Silicate Load and Exfoliated Nanolayers Adsorbed on Polymer Chains"

_polymers, 2023, doi:10.3390/polym15061413_

Round 1

Reviewer 1 Report

In this work, the authors have proposed a novel polymer nanocomposite chemistry with improved temperature, salt, and shear resistance, presenting a potential opportunity for its application in enhanced oil recovery (EOR) in high temperature and high salinity reservoirs. While the research is interesting, I believe the manuscript, in its current state, is not meeting up to the standards of MDPI polymers. There are several reasons that have made me come to this conclusion. I have presented them below for the benefit of the authors. However, these concerns/questions must be addressed before it is published in any high-impact scientific journal.

After careful reading, I would like to offer my comments as follows.

1.     The authors have provided an exhaustive introduction with relevant references. This is great to establish a background for the research work. However, in the current scenario, the novelty of the work is lost and the purpose of undertaking the present research is unclear. There seems to be a lack of logical flow. 

To help make the distinction clear and bring out the work’s significance, it would be helpful to highlight the novelty using the proposed layout: (a) key highlights of the previous studies, (b) crucial points lacking in those studies, (c) importance of the missing pieces from a big-picture perspective, (d) how does the current study help to address those to advance the understanding, and (e) how does the current findings help the field – future scope and applications of the observations. This will help the reader to appreciate the current study and clarify the impact of the work.

2.     I have a very big concern regarding the in-situ polymerization of the polymer nanocomposite. The authors have gravely missed to explain the polymerization mechanism and how they control the molecular weights for each sample type. The differences among the samples based on their molecular weights can be found in Table 3 in the manuscript. Since comparisons are made among different samples, it is essential to draw conclusions simply based on the effect of filler (O-MMt) concentration alone. If there are variations in molecular weights among samples in several hundred thousands’, one cannot disregard the effects resulting from the molecular weight of the polymer itself. This makes the analysis complicated and casts a doubt on the performance conclusions drawn. The authors are required to rectify this issue and provide data to refute that the molecular weights are not interfering in their observations. 

Additionally, the authors are expected to have run the polymerization multiple times for each sample type and to provide standard deviations for the molecular weights reported in Table 3. It will provide the reader with a sense of degree of control over the reaction and reproducibility.

3.     Building on point 2, let’s focus on the viscosity-related plots presented in this study.

a)     Firstly, why are there some plots (like Table 3Figure 14, and Figure 15) without the ASD/0.5 wt% O-MMt case? Can the authors supplement the data for the same in the main body of the manuscript along with the other cases? 

Moreover, all the plots are lacking standard deviations for each data point. Can the authors comment on how many repeats were run for each test? If none, could they please perform multiple runs and provide standard deviations? This is an important step towards data repeatability, reproducibility, and data interpretation.

b)    In Figure 11Figure 12, and Figure 13, can the authors explain why ASD/0.5 wt% O-MMt and ASD/2.0 wt% O-MMt less viscous than the pure polymer ASD? There are certainly huge molecular weight differences among them. In addition, can the authors comment on the trends observed with increasing concentration of O-MMt in each study case? 

c)     In Figure 13, why do we observe such a sudden initial drop for pure polymer ASD, meanwhile other cases show a gradual drop with increasing salt concentration? 

d)    Until Figure 13, all the viscosity-related plots showed that the ASD/2.0 wt% O-MMt was less viscous than pure ASD. However, in Figure 14, the ASD/2.0 wt% O-MMt sample now exhibits higher apparent viscosities with shear rate than pure polymer. Can the authors comment on this observation and note any discrepancy if present? Is there any sensitivity to shear rate for these filled composite systems?

e)     In Figure 15, why has the crossover point (or characteristic frequency) dramatically shifted to lower frequencies for ASD/1.0 wt% O-MMt sample, but only marginally for ASD/2.0 wt% O-MMt with respect to the pure ASD sample? I don’t believe this behavior can be simply explained by O-MMt concentration, knowing the fact that molecular weight differences among the samples are large. 

f)     Finally, one minor comment. Please include the R-squared values for the linear fits in Figure 10. Moreover, please include the ηsp/or lnηr/c plots for other polymer samples (ASD/0.5 wt% O-MMt, ASD/1.0 wt% O-MMt, and ASD/2.0 wt% O-MMt) in the supplementary information.

In conclusion to this part, it feels as if the authors have simply recorded observations in the Results section but overlooked some of these trends/discrepancies which could have crucial implications on the O-MMt – polymer interaction. Is there a sweet-spot for O-MMt loading concentration to get an optimal performance? How can this be explained considering each of the sample system has largely varying molecular weights?

4.     In Figure 16, it would have been great if the authors could have provided EOR experiment plots for pure ASD, ASD/0.5 wt% O-MMt, and ASD/2.0 wt% O-MMt cases. If there have been comparisons made for all other characterizations, why miss out during the most crucial application test? The authors are expected to provide these results to have a full understanding of the system and highlight the effect of O-MMt concentration on EOR performance. 

5.     In the TEM images supplied in Figure 8, it is not clear as to how to interpret images (c) and (d). The figure caption reads well-dispersed, but the images show some clustering (dark, dense areas of the TEM image). Can the authors clarify this and provide better TEM images to support their claims? Additionally, it would be helpful to provide TEM images for pure ASD, ASD/0.5 wt% O-MMt, and ASD/2.0 wt% O-MMt cases in the supporting information for reader’s convenience.

6.     Similar to point 5, the SEM images in Figure 7 are not conclusive enough to explain differences between pure ASD and O-MMt-loaded ASD composites. The authors should consider using complementary techniques like EDS to highlight areas where O-MMt is present. This would be more accurate representation of how well-dispersed the O-MMt is for one system over the other. Since O-MMt is rich is Si-element, I believe it should be fairly possible to use EDS to discern the differences among different polymer systems. 

Again, the authors are expected to provide SEM/EDS images for the rest of composite systems (in addition to the ones in Figure 7) – ASD/0.5 wt% O-MMt and ASD/2.0 wt% O-MMt.

7.     With regards to the XRD characterization of the polymer composite system, what type of diffraction geometry was employed? How were the samples prepared? Were they powdered or spun-coated on silicon wafers? Please provide appropriate details in the Methods section.

Why is it that in none of the composite cases (0 wt%, 0.5 wt%, 1.0 wt%, and 2.0 wt%) we do not observe any diffraction peaks from O-MMt (Figure 6), especially in high-loaded samples? Is it safe to assume that the concentrations are too small to be detected by the X-ray detector? Have the authors tried preparing very high-loaded samples of O-MMt polymer nanocomposite to test if O-MMt is detected? It is challenging to accept that simply because we don’t see diffraction peaks for O-MMt, they are well exfoliated and well dispersed. The X-ray detector may not be sensitive enough to pick up signals from such low-concentrated samples. Can the authors comment on the same?

8.     With respect to FTIR spectroscopy, the authors must provide appropriate references for the peak assignments used to explain the FTIR spectra in Figure 4 for cross-reference purposes. Furthermore, why do we see same spectral signatures in the band region 400 – 1000 cm-1 for ASD and ASD/1.0 wt% O-MMt? As the spectral signatures are peculiar to Si-O-Si bending and Si-O-Al stretching vibrations, they should have shown up only in the O-MMt loaded cases. However, we see those in pure ASD as well. Can the authors comment on the same? 

Please also provide the FTIR spectra for ASD/0.5 wt% O-MMt and ASD/2.0 wt% O-MMt systems in the supplementary information. 

9.     Lastly, have the authors provided an existing mechanism for O-MMt nanolayer adsorption or proposed one? In the former case, please provide suitable references to allow readers to cross-refer. In the latter case, please make it clear in the section and figure caption that this is a proposed mechanism. Additionally, while proposing a mechanism, it is imperative to provide references on whose foundation the mechanism was drafted. Please provide any (if available) for cross-reference purposes.

Author Response

Point-by-point response to the reviewers' comments

Polymers

Manuscript ID: polymers-2163532

Title: Preparation and properties of water-soluble copolymer nanocomposites with strongly adsorbed exfoliated montmorillonite nanolayers

Dear editor Meredith Wei and reviewers, thanks very much for taking your time to review this manuscript. I really appreciate all your comments and suggestions! According to your suggestions, we have made the following revisions on this manuscript:

Response to reviewer #1:

We really appreciate you for your carefulness and conscientiousness. Your suggestions are really valuable and helpful for revising and improving our paper.

Comment: 1

The authors have provided an exhaustive introduction with relevant references. This is great to establish a background for the research work. However, in the current scenario, the novelty of the work is lost and the purpose of undertaking the present research is unclear. There seems to be a lack of logical flow.

To help make the distinction clear and bring out the work’s significance, it would be helpful to highlight the novelty using the proposed layout: (a) key highlights of the previous studies, (b) crucial points lacking in those studies, (c) importance of the missing pieces from a big-picture perspective, (d) how does the current study help to address those to advance the understanding, and (e) how does the current findings help the field – future scope and applications of the observations. This will help the reader to appreciate the current study and clarify the impact of the work.

Response: Thank you very much for your insightful comment. We have revised the manuscript to highlight the novelty and the purpose of the work. Firstly, the water-soluble polymer nanocomposite with well-exfoliated and well-dispersed O-MMt nanolayers strongly adsorbed on polymer chains were prepared. Secondly, the water-soluble polymer nanocomposite (ASD/O-MMt) exhibited good temperature, salt, and shear resistance. Thirdly, the water-soluble polymer nanocomposite demonstrates significant potential for oil-recovery applications in high temperature and high salt reservoir.

Comment: 2

I have a very big concern regarding the in-situ polymerization of the polymer nanocomposite. The authors have gravely missed to explain the polymerization mechanism and how they control the molecular weights for each sample type. The differences among the samples based on their molecular weights can be found in Table 3 in the manuscript. Since comparisons are made among different samples, it is essential to draw conclusions simply based on the effect of filler (O-MMt) concentration alone. If there are variations in molecular weights among samples in several hundred thousands’, one cannot disregard the effects resulting from the molecular weight of the polymer itself. This makes the analysis complicated and casts a doubt on the performance conclusions drawn. The authors are required to rectify this issue and provide data to refute that the molecular weights are not interfering in their observations.

Additionally, the authors are expected to have run the polymerization multiple times for each sample type and to provide standard deviations for the molecular weights reported in Table 3. It will provide the reader with a sense of degree of control over the reaction and reproducibility.

Response: Thank you for your positive comments, the mechanism of in-situ polymerization has been added to the manuscript. A large number of active monomers intercalated into the interlayers of O-MMt in the process of in-situ polymerization, thus improving the polymerization activity and molecular weight of the polymer nanocomposites. However, when O-MMt agglomerates, the polymerization activity and the molecular weight of the polymer will be reduced, such as ASD/0.5 wt% O-MMt and ASD/2.0 wt% O-MMt. In the process of preparing polymer, the other conditions are consistent except for the different amount of O-MMt. All polymerization reactions have been repeated more than three times, and the samples obtained each time can have the same properties, which can ensure the reproducibility. The standard deviations for ASD, ASD/0.5 wt% O-MMt, ASD/1.0 wt% O-MMt and ASD/2.0 wt% O-MMt are about 1.39, 1.63, 1.27 and 1.98.

Comment: 3

Building on point 2, let’s focus on the viscosity-related plots presented in this study.

  1. a) Firstly, why are there some plots (like Table 3, Figure 14, and Figure 15) without the ASD/0.5 wt% O-MMt case? Can the authors supplement the data for the same in the main body of the manuscript along with the other cases?

Moreover, all the plots are lacking standard deviations for each data point. Can the authors comment on how many repeats were run for each test? If none, could they please perform multiple runs and provide standard deviations? This is an important step towards data repeatability, reproducibility, and data interpretation.

Response: Thank you very much for your advice, the curves of ASD/0.5 wt% O-MMt has been added to the main body of the manuscript along with the other cases and all the plots are repeated three times for each test.

  1. b) In Figure 11, Figure 12, and Figure 13, can the authors explain why ASD/0.5 wt% O-MMt and ASD/2.0 wt% O-MMt less viscous than the pure polymer ASD? There are certainly huge molecular weight differences among them. In addition, can the authors comment on the trends observed with increasing concentration of O-MMt in each study case?

Response: Thank you so much for your careful check. Due to the addition of O-MMt changes the polarity of the polymer side group and the morphology of the polymer molecular chain in the aqueous solution, resulting in the decrease of the apparent viscosity of the aqueous solution. When the concentration of O-MMt is 1.0 wt%, the O-MMt nanolayers are able to be well-exfoliated and well-dispersed in the polymer matrix. So there is a strong adsorption interaction between O-MMt nanolayers and polymer chains. The apparent viscosity of polymer nanocomposite aqueous solution is the largest, which can better resist high temperature and salt.

  1. c) In Figure 13, why do we observe such a sudden initial drop for pure polymer ASD, meanwhile other cases show a gradual drop with increasing salt concentration?

Response: Thank you for your comment. The polymer chains of ASD rapidly curls and contracts, resulting in a sudden drop in apparent viscosity.

  1. d) Until Figure 13, all the viscosity-related plots showed that the ASD/2.0 wt% O-MMt was less viscous than pure ASD. However, in Figure 14, the ASD/2.0 wt% O-MMt sample now exhibits higher apparent viscosities with shear rate than pure polymer. Can the authors comment on this observation and note any discrepancy if present? Is there any sensitivity to shear rate for these filled composite systems?

Response: Thank you for pointing out this problem in manuscript. During the continuous shear process, due to the addition of 2 wt% O-MMt, the friction force and shear stress increase, thus increasing the apparent viscosity.

  1. e) In Figure 15, why has the crossover point (or characteristic frequency) dramatically shifted to lower frequencies for ASD/1.0 wt% O-MMt sample, but only marginally for ASD/2.0 wt% O-MMt with respect to the pure ASD sample? I don’t believe this behavior can be simply explained by O-MMt concentration, knowing the fact that molecular weight differences among the samples are large.

Response: Thank you very much for your advice, the molecular weight of polymer can affect the viscoelasticity. Interestingly, there is a strong adsorption interaction between the well-exfoliated and well-dispersed O-MMt nanolayers and polymer chains which also affect the viscoelasticity of the polymer.

  1. f) Finally, one minor comment. Please include the R-squared values for the linear fits in Figure 10. Moreover, please include the ηsp/c or lnηr/c plots for other polymer samples (ASD/0.5 wt% O-MMt, ASD/1.0 wt% O-MMt, and ASD/2.0 wt% O-MMt) in the supplementary information.

Response: Thank you for your comments. The R-squared values for the linear fits in Figure 10 is 0.97. The ηsp/c or lnηr/c plots for other polymer samples (ASD/0.5 wt% O-MMt, ASD/1.0 wt% O-MMt, and ASD/2.0 wt% O-MMt) will be added in the supplementary information.

In conclusion to this part, it feels as if the authors have simply recorded observations in the Results section but overlooked some of these trends/discrepancies which could have crucial implications on the O-MMt – polymer interaction. Is there a sweet-spot for O-MMt loading concentration to get an optimal performance? How can this be explained considering each of the sample system has largely varying molecular weights?

Response: Thank you very much for your advice. When the amount of O-MMt is 1 wt%, the O-MMt nanolayers are able to be well-exfoliated and well-dispersed in the polymer matrix and there is a strong adsorption interaction between the well-exfoliated and well-dispersed O-MMt nanolayers and polymer chains which also affect the property of polymer.

  1. In Figure 16, it would have been great if the authors could have provided EOR experiment plots for pure ASD, ASD/0.5 wt% O-MMt, and ASD/2.0 wt% O-MMt cases. If there have been comparisons made for all other characterizations, why miss out during the most crucial application test? The authors are expected to provide these results to have a full understanding of the system and highlight the effect of O-MMt concentration on EOR performance.

Response: Thank you for your comments. By comparing all other characterizations among the samples (ASD, ASD/0.5 wt% O-MMt, ASD/1.0 wt% O-MMt, ASD/2.0 wt% O-MMt), the ASD/1.0 wt% O-MMt was the best. Therefore, the EOR experiment was only tested of ASD/1.0 wt% O-MMt. The others will be added in the supplementary information.

  1. In the TEM images supplied in Figure 8, it is not clear as to how to interpret images (c) and (d). The figure caption reads well-dispersed, but the images show some clustering (dark, dense areas of the TEM image). Can the authors clarify this and provide better TEM images to support their claims? Additionally, it would be helpful to provide TEM images for pure ASD, ASD/0.5 wt% O-MMt, and ASD/2.0 wt% O-MMt cases in the supporting information for reader’s convenience.

Response: Thank you very much for your advice. Figure 8 (c) and (d) showed the well-exfoliated O-MMt nanolayers well-dispersed in polymer matrix and the dark areas was the polymer matrix. The TEM images of ASD, ASD/0.5 wt% O-MMt, and ASD/2.0 wt% O-MMt will be added in the supplementary information.

  1. Similar to point 5, the SEM images in Figure 7 are not conclusive enough to explain differences between pure ASD and O-MMt-loaded ASD composites. The authors should consider using complementary techniques like EDS to highlight areas where O-MMt is present. This would be more accurate representation of how well-dispersed the O-MMt is for one system over the other. Since O-MMt is rich is Si-element, I believe it should be fairly possible to use EDS to discern the differences among different polymer systems.

Again, the authors are expected to provide SEM/EDS images for the rest of composite systems (in addition to the ones in Figure 7) – ASD/0.5 wt% O-MMt and ASD/2.0 wt% O-MMt.

Response: Thank you very much for your advice, the EDS has been added to the main body of the manuscript.

  1. With regards to the XRD characterization of the polymer composite system, what type of diffraction geometry was employed? How were the samples prepared? Were they powdered or spun-coated on silicon wafers? Please provide appropriate details in the Methods section.

Response: Thank you for your rigorous consideration. The samples were powder and appropriate details has been revised in main body of the manuscript.

Why is it that in none of the composite cases (0 wt%, 0.5 wt%, 1.0 wt%, and 2.0 wt%) we do not observe any diffraction peaks from O-MMt (Figure 6), especially in high-loaded samples? Is it safe to assume that the concentrations are too small to be detected by the X-ray detector? Have the authors tried preparing very high-loaded samples of O-MMt polymer nanocomposite to test if O-MMt is detected? It is challenging to accept that simply because we don’t see diffraction peaks for O-MMt, they are well exfoliated and well dispersed. The X-ray detector may not be sensitive enough to pick up signals from such low-concentrated samples. Can the authors comment on the same?

Response: We gratefully thanks for the precious time the reviewer spent making constructive remarks. XRD analysis has been revised in main body of the manuscript. ASD is an amorphous material and the polymer chains inserted into the interlayer of O-MMt during the polymerization process. And the polymerization reaction is exothermic, these heat speeds up the exfoliated of O-MMt nanolayers dispersed in the polymer matrix. The high-loaded samples of O-MMt polymer nanocomposite will be studied in the future.

  1. With respect to FTIR spectroscopy, the authors must provide appropriate references for the peak assignments used to explain the FTIR spectra in Figure 4 for cross-reference purposes. Furthermore, why do we see same spectral signatures in the band region 400 – 1000 cm-1 for ASD and ASD/1.0 wt% O-MMt? As the spectral signatures are peculiar to Si-O-Si bending and Si-O-Al stretching vibrations, they should have shown up only in the O-MMt loaded cases. However, we see those in pure ASD as well. Can the authors comment on the same?

Please also provide the FTIR spectra for ASD/0.5 wt% O-MMt and ASD/2.0 wt% O-MMt systems in the supplementary information.

Response: Thank you so much for your careful check. The references related to FTIR analysis has been revised in main body of the manuscript. Due to the small amount of O-MMt, the Si-O-Si bending and Si-O-Al stretching vibrations may overlap with the peak of ASD. And the FTIR spectra of ASD/0.5 wt% O-MMt and ASD/2.0 wt% O-MMt was provided in the supplementary information.

  1. Lastly, have the authors provided an existing mechanism for O-MMt nanolayer adsorption or proposed one? In the former case, please provide suitable references to allow readers to cross-refer. In the latter case, please make it clear in the section and figure caption that this is a proposed mechanism. Additionally, while proposing a mechanism, it is imperative to provide references on whose foundation the mechanism was drafted. Please provide any (if available) for cross-reference purposes.

Response: Thank you for your comments. The mechanism for O-MMt nanolayer adsorption is proposed, and the reference has been added in main body of the manuscript.

Reviewer 2 Report

Paper needs editing of English.

In title what is strongly adsorbed?

How was this strength of abrorption determined?

The title needs to be modified properly.

Otherwise explain the strength of absorption.

How does the microscopic image indicate exfoliation? Show a bigger image to see the overall picture.

Author Response

Point-by-point response to the reviewers' comments

Polymers

Manuscript ID: polymers-2163532

Title: Preparation and properties of water-soluble copolymer nanocomposites with strongly adsorbed exfoliated montmorillonite nanolayers

Dear editor Meredith Wei and reviewers, thanks very much for taking your time to review this manuscript. I really appreciate all your comments and suggestions! According to your suggestions, we have made the following revisions on this manuscript:

Response to reviewer #2:

We gratefully thank for the precious time the reviewer spent making constructive remarks. Your suggestions are really valuable and helpful for revising and improving our paper.

Comment: 1

In title what is strongly adsorbed?

Response: Thank you for your positive comments, the strong adsorption refers to the strong interaction force between the O-MMt nanolayers and the polymer molecular chains, which mainly includes hydrogen bond, intermolecular force and van der Waals force.

Comment: 2

How was this strength of abrorption determined?

Response: Thank you for pointing out this problem in manuscript. From the SEM images, the polymer surface is rough due to the strong adsorption of O-MMt nanolayers. In addition, the theoretical calculation shows that the strong adsorption energy is about -13032 kcal/mol.

Comment: 3

The title needs to be modified properly. Otherwise explain the strength of absorption.

Response: Thank you very much for your advice. The title has been revised. The strength of absorption can be calculated by theoretical calculation.

Comment: 4

How does the microscopic image indicate exfoliation? Show a bigger image to see the overall picture.

Response: Thank you for your positive comments. From the TEM, the exfoliated O-MMt nanolayer is single and dispersed in the polymer matrix.

Reviewer 3 Report

Dear Editor and Authors,

I have reviewed this paper with close inspection. The topic of the present publication is interesting but I have the following comments for the authors prior its publication: 

1. Moderate corrections in English grammar and syntax are required.

2. The XRD section needs further description and details. The peaks must be described in Armstrong. Moreover, the clay layers may exfoliate but it is not possible to lose their crystallinity. Please revise this part. Please modify lines 334-335 accordingly. 

3. The verification of the mineral's content in the particles based on TGA results must be included. Equations used in the following reference could be used: https://doi.org/10.1016/j.jddst.2019.05.001

4. Separate TGA and DTG curves.

5. Provide 3 repetitions in the viscosity experiments. The results do not present a logical sequence related to the concentration of O-MMt. How do the authors describe this phenomenon? 

Author Response

Point-by-point response to the reviewers' comments

Polymers

Manuscript ID: polymers-2163532

Title: Preparation and properties of water-soluble copolymer nanocomposites with strongly adsorbed exfoliated montmorillonite nanolayers

Dear editor Meredith Wei and reviewers, thanks very much for taking your time to review this manuscript. I really appreciate all your comments and suggestions! According to your suggestions, we have made the following revisions on this manuscript:

Response to reviewer #3:

We gratefully appreciate for your valuable comments. Your suggestions are really valuable and helpful for revising and improving our paper.

Comment: 1

Moderate corrections in English grammar and syntax are required.

Response: Thank you for your comments. We apologize for the poor language of our manuscript. We have now worked on both language and readability and have also involved native English speakers for language corrections. We really hope that the flow and language level have been substantially improved.

Comment: 2

The XRD section needs further description and details. The peaks must be described in Armstrong. Moreover, the clay layers may exfoliate but it is not possible to lose their crystallinity. Please revise this part. Please modify lines 334-335 accordingly.

Response: We feel sorry for the inconvenience brought to the reviewer. The XRD section and lines 334-335 have been revised.

Comment: 3

The verification of the mineral's content in the particles based on TGA results must be included. Equations used in the following reference could be used: https://doi.org/10.1016/j.jddst.2019.05.001

Response: Thank you very much for your advice. TGA has been revised and the reference has been added.

Comment: 4

Separate TGA and DTG curves.

Response: Thank you for your comments. TGA and DTG curves has been revised.

Comment: 5

Provide 3 repetitions in the viscosity experiments. The results do not present a logical sequence related to the concentration of O-MMt. How do the authors describe this phenomenon?

Response: Thank you very much for your advice. The polymer nanocomposites with excellent performance cannot be obtained by adding too much or too little O-MMt. Only when the amount of O-MMt is 1.0 wt%, the O-MMt nanolayers can be well exfoliated and dispersed in the polymer matrix and have strong adsorption with the polymer molecular chains. At this time, the polymer nanocomposites obtained have excellent performance.

Round 2

Reviewer 1 Report

I thank the authors for making an effort to address the concerns. However, before final publishing, I would still like the authors, for readers convenience, to include/incorporate standard deviations (or error bars) in the Table 3 and in the appropriate figure plots (viscosity and rheological plots), and present R^2 information for the linear fits in Figure 10 (and corresponding supplementary plots). 

Author Response

Point-by-point response to the reviewers' comments

Polymers

Manuscript ID: polymers-2163532

Title: Preparation and properties of water-soluble copolymer nanocomposites with strongly adsorbed exfoliated montmorillonite nanolayers

Dear editor Meredith Wei and reviewers, thanks very much for taking your time to review this manuscript. I really appreciate all your comments and suggestions!

Response to reviewer #1:

We gratefully appreciate for your valuable comments. Your suggestions are really valuable and helpful for revising and improving our paper.

Comment: 1

I thank the authors for making an effort to address the concerns. However, before final publishing, I would still like the authors, for readers convenience, to include/incorporate standard deviations (or error bars) in the Table 3 and in the appropriate figure plots (viscosity and rheological plots), and present R^2 information for the linear fits in Figure 10 (and corresponding supplementary plots).

Response: Thank you very much for your advice. The standard deviations have been added in Table 3. And the R^2 information for the linear fits in Figure 10 (and corresponding supplementary plots) have been added.

Reviewer 3 Report

Dear Editor & authors,

I have no further comments for the approval of the present manuscript!

Author Response

Point-by-point response to the reviewers' comments

Polymers

Manuscript ID: polymers-2163532

Title: Preparation and properties of water-soluble copolymer nanocomposites with strongly adsorbed exfoliated montmorillonite nanolayers

Dear editor Meredith Wei and reviewers, thanks very much for taking your time to review this manuscript. I really appreciate all your comments and suggestions!

We gratefully appreciate for your valuable comments. Your suggestions are really valuable and helpful for revising and improving our paper.